# Hospital-Treated Infections and Increased Risk of Two EBV-Related Malignancies: A Nested Case-Control Study

**DOI:** 10.3390/cancers14153804

**Published:** 2022-08-05

**Authors:** Yanping Yang, Li Yin, Qianwei Liu, Jiangwei Sun, Hans-Olov Adami, Weimin Ye, Zhe Zhang, Fang Fang

**Affiliations:** 1Department of Otolaryngology-Head & Neck Surgery, First Affiliated Hospital of Guangxi Medical University, Nanning 530021, China; 2Key Laboratory of Early Prevention and Treatment for Regional High-Frequency Tumor (Guangxi Medical University), Ministry of Education/Guangxi Key Laboratory of High-Incidence-Tumor Prevention & Treatment (Guangxi Medical University), Nanning 530021, China; 3Department of Medical Epidemiology and Biostatistics, Karolinska Institutet, 17177 Stockholm, Sweden; 4Institute of Environmental Medicine, Karolinska Institutet, 17177 Stockholm, Sweden; 5Clinical Effectiveness Group, Institute of Health and Society, University of Oslo, 0315 Oslo, Norway; 6Department of Epidemiology and Health Statistics & Key Laboratory of Ministry of Education for Gastrointestinal Cancer, Fujian Medical University, Fuzhou 350005, China

**Keywords:** infection, Epstein-Barr virus, malignancy, risk, nested case-control study

## Abstract

**Simple Summary:**

Epstein–Barr virus (EBV) is associated with the risk of several human malignancies, including Hodgkin’s lymphoma (HL) and nasopharyngeal carcinoma (NPC). However, as EBV infection is widely spread across human populations, whereas only a small proportion of the infected individuals eventually develop such malignancies, additional component causes are likely needed in the development of EBV-related malignancies. We performed a national register-based study in Sweden to examine the role of infections that required hospital treatment on the subsequent risk of HL and NPC. We found that previous hospital-treated infections were associated with a higher risk of HL and NPC and that a positive association was observed for both bacterial and viral infections, especially for respiratory and skin infections. A dose-response relationship was also noted between the number of infections and the risk of HL. These findings suggest that infectious events might contribute to the carcinogenesis of malignancies potentially related to EBV.

**Abstract:**

Background: To assess the association of hospital-treated infections with the subsequent risk of two Epstein-Barr virus (EBV)-related malignancies, namely Hodgkin’s lymphoma (HL) and nasopharyngeal carcinoma (NPC). Methods: We performed a nested case-control study based on several national registers in Sweden. Cases were individuals newly diagnosed with HL or NPC during 1994–2016 in Sweden, according to the Swedish Cancer Register. For each case, we randomly selected five controls individually matched to the case on sex and year of birth from the general Swedish population. Hospital-treated infections (i.e., infections requiring either inpatient or outpatient hospital care) were identified from the Swedish Patient Register. Conditional logistic regression was used to estimate odds ratios (ORs) and 95% confidence intervals (CIs) of HL and NPC, in relation to hospital-treated infections, after adjustment for age, sex, calendar period, educational achievement, and region of residence. Results: The study included a total of 890 cases of HL and 306 cases of NPC. A hospital-treated infection three years ago or earlier was associated with a higher risk of HL (OR = 1.49, 95%CI: 1.26–1.75) as well as NPC (OR = 1.36; 95%CI: 1.01–1.83). The positive association was noted for both bacterial and viral infections and primarily for respiratory and skin infections. A monotonous dose-response relationship was found between a number of hospital-treated infections and the risk of HL (*p* = 0.02) but less compelling for NPC (*p* = 0.06). Using a 5-year lag time rendered similar results (OR = 1.43, 95%CI: 1.21–1.70 for HL; OR = 1.43, 95%CI: 1.05–1.95 for NPC). Conclusions: These findings suggest that infections requiring hospital treatment might contribute to the carcinogenesis of malignancies potentially related to EBV.

## 1. Introduction

Infection plays an established causal role in several human malignancies, with a growing number of pathogens recognized as oncogenic [1]. Epstein–Barr virus (EBV), also known as human herpes virus type 4 (HHV4), is, for instance, associated with the risk of Hodgkin’s lymphoma (HL) [2], nasopharyngeal carcinoma (NPC) [3], non-Hodgkin’s lymphoma, especially Burkitt lymphoma, and T/NK-cell lymphomas and lymphoproliferative diseases [4,5]. Many oncogenic pathogens, including EBV, are widely spread across human populations and often acquired relatively early in life, whereas only a small proportion of the infected individuals eventually develop the malignancies potentially attributable to them [6]. Hence, additional component causes are needed for such infection-related malignancies. In the case of NPC, studies have suggested the interaction between genetic risk factors and EBV infection [6] as well as between lifestyle factors and EBV infection [7]. Similar efforts have also been made in HL [8,9].

The substantial global variance in infections with an oncogenic pathogen, as well as the burden of the related malignancies, adds another layer to the complexity of understanding the etiologies of infection-related malignancies [10]. Related to this, the proportion of cases attributable to the specific pathogen varies substantially between regions and other factors. For instance, EBV-attributable cases, in which EBV viral DNA/RNA or viral gene expression can be demonstrated in the tumor tissue, vary between types of malignancies and by region and age [11]. Indeed, EBV is identified among almost all cases of NPC in regions with a high or intermediate incidence but among 80% of cases of NPC in regions with a low incidence [11]. Similarly, the prevalence of EBV is 95% among cases of Burkitt lymphoma in endemic areas, 50% in intermediate areas, but only 20% in non-endemic areas [11]. Finally, across the globe, the prevalence of EBV is estimated to be 62% among children (below 15) with HL, 30% among young adults (15–54 years) with HL, and 55% among elderly people (>55 years) with HL [11]. Additional risk factors, independently of or jointly with EBV infection, might, therefore, contribute to the cases of EBV-related malignancies not defined as EBV attributable.

One underexplored area is the role of infections, especially the ones not due to oncogenic pathogens, on the risk of malignant diseases in general and infection-related malignancies specifically [12]. Inflammation after infections, especially severe infections, may make normal cells more susceptible to carcinogenesis, whereas compromised host immunity following severe infections might lead to aberrant immunosurveillance in the clearance of oncogenic organisms and cancer cells [13]. To this end, we performed a national register-based study in Sweden to examine the role of infections that required hospital treatment on the risk of two EBV-related malignancies, namely HL and NPC. Our hypothesis is that hospital-treated infections, regardless of the specific pathogens involved, contribute to the carcinogenesis of HL and NPC. We decided not to study other EBV-related malignancies, such as Burkitt lymphoma and gastric cancer, because of the presumably low attribution of EBV to these malignancies in the Swedish population.

## 2. Materials and Methods

### 2.1. Study Design

We first performed a cohort study including all Swedes born since 1900 in Sweden with both parents also born in Sweden, according to the Swedish Total Population Register (*N* = 12,275,551). We followed these individuals from 1 January 1994 until a diagnosis of HL or NPC, emigration out of Sweden, death, or 31 December 2016, whichever occurred first, through cross-linking the cohort to the Swedish Cancer Register and the Total Population Register, using the individually unique national registration numbers. The Swedish Cancer Register has collected information on all newly diagnosed malignancies in Sweden since 1958 [14]. We identified incident cases of HL and NPC during the follow-up, using the 7th Swedish revision of the International Classification of Disease (ICD) codes (ICD-7 code 201 for HL and 146 for NPC).

We then conducted a nested case-control study within the above study base to assess the association of hospital-treated infections with the subsequent risk of HL and NPC. Using the method of incidence density sampling [15], for each case of HL and NPC, we randomly selected five controls, individually matched to the case by sex and year of birth, from the study base. We used the date of cancer diagnosis for the case and the date of selection for the controls as the index date. The controls had to be alive and free of HL and NPC on the index date.

### 2.2. Hospital-Treated Infections

We identified hospital-treated infections requiring either inpatient or outpatient care, before the index date for the cases and controls, by linking all participants in the nested case-control study to the Swedish Patient Register. This register collects nationwide data on inpatient hospital care records from 1987 onward and outpatient hospital care records from 2001 onward, including information on the date of a hospital visit, primary and secondary diagnoses, and different procedures in Sweden. The diagnoses are recorded using different versions of the Swedish revisions of ICD codes. We identified all hospital visits with a diagnosis of infection, either as the primary diagnosis or as a secondary diagnosis, as hospital-treated infections. Please see Appendix A for the ICD codes used to identify infections. We first studied any infection and then studied infections by type (i.e., viral, bacterial, or other) and site (i.e., central nervous system, gastrointestinal, genitourinary, respiratory, or skin) of infection. To understand the potential high-risk time window, we also performed an analysis by age at infection (<40, 40–59.9, or ≥60 years), and to understand the potential dose–response relationship, we studied the number of infection events (0, 1, or ≥2).

### 2.3. Covariates

We linked participants in the nested case-control study to the Swedish Longitudinal Integrated Database for Health Insurance and Labor to obtain data on the area of residence (Southern, Central, or Northern Sweden) and educational attainment (0–9 years, 10–12 years, ≥13 years, or “missing”) before the index date. Family history of HL or NPC was defined as a diagnosis of the disease among the first-degree relatives (i.e., biological parents and full siblings) of the cases and controls. We identified the parents and siblings of the cases and controls through the Swedish Multi-Generation Register, which contains largely complete information on familial links for Swedish residents born since 1932 [16].

### 2.4. Statistical Analysis

We used conditional logistic regression to estimate the odds ratio (OR) and 95% confidence interval (CI) of HL and NPC in relation to hospital-treated infections. ORs obtained from a nested case-control study represent a relative risk estimate of the underlying study base [17]. In addition to matching variables, we adjusted the analyses for calendar period, area of residence, and educational attainment. To reduce any impact of surveillance bias (i.e., greater-than-expected surveillance of HL or NPC following infections) and reverse causation (i.e., infections as a secondary event to pre-clinical HL or NPC), we used a lag time of three years in the main analysis and excluded from the analysis infections during the three years before the index date. We performed the analyses for any infection first and by type, site, age, and the number of infections subsequently. To assess whether the association of hospital-treated infections with the risk of HL and NPC would differ between males and females or between younger and older individuals, we also stratified the analysis of any hospital-treated infection by sex and age (<60 or ≥60 years). To assess the impact of family history, in a sensitivity analysis, we restricted the analysis to individuals without a family history of HL or NPC. Further, as infectious mononucleosis is also related to EBV infection [18] and has been shown to be related to the risk of HL [19], we excluded infectious mononucleosis from the definition of hospital-treated infections in another sensitivity analysis. Finally, to assess the influence of lag time, we also performed a sensitivity analysis where hospital-treated infections experienced during the five years before the index date were excluded.

Data management and analyses were performed using SAS version 9.4 (SAS Institute Inc., Cary, NC, USA) and R version 3.6.0. A two-sided *p* ≤ 0.05 was considered statistically significant. We did not adjust for the multiplicity of statistical tests, as adopting a top-down approach, the main hypothesis of increased risk of HL and NPC in relation to hospital-treated infections consisted of only two tests.

## 3. Results

In the present study, we included 890 cases of HL and 306 cases of NPC, together with their individually matched controls (Table 1). Among the cases, one individual had a diagnosis of both HL and NPC. The median age at diagnosis was 71.2 for patients with HL and 66.5 for patients with NPC. For both malignancies, there were more male than female patients.

After excluding infections diagnosed during three years before the index date, a hospital-treated infection was associated with an increased risk of both HL (OR = 1.49; 95%CI: 1.26–1.75) and NPC (OR = 1.36; 95%CI: 1.01–1.83) (Table 2). For HL, a positive association of similar magnitude was observed for both bacterial and viral infections, as well as for respiratory and skin infections. No association was noted for gastrointestinal or genitourinary infections. For NPC, the subgroup analyses were not statistically significant. A dose–response relationship was noted between the number of hospital-treated infections and the risk of HL (*p* = 0.02), with an 89% increased risk of NPC following three or more infections. A less monotonous and statistically non-significant (*p* = 0.06) dose-response relationship was found for NPC.

The association of hospital-treated infections with a higher risk of HL and NPC was noted among males and females as well as among younger and older individuals, although some of the estimates were not statistically significant for NPC (Table 3). Because of the small number of individuals with a positive family history, excluding these individuals from the analysis had little impact on the results (any hospital-treated infections: OR = 1.49, 95%CI: 1.26–1.75 for HL, and OR = 1.36, 95%CI: 1.01–1.83 for NPC). Among the cases and controls with a history of hospital-treated infections, three controls had a diagnosis of infectious mononucleosis. Excluding these individuals from the analysis did not change the results (any hospital-treated infections: OR = 1.49, 95%CI: 1.27–1.76 for HL; and OR = 1.36, 95%CI: 1.01–1.83 for NPC). Finally, using a lag-time of five years rendered similar results (any hospital-treated infections: OR = 1.43, 95%CI: 1.21–1.70 for HL, and OR = 1.43, 95%CI: 1.05–1.95 for NPC).

## 4. Discussion

In this large population-based study, hospital-treated infections three or five years ago or earlier were associated with a higher subsequent risk of two EBV-related malignancies, HL and NPC. A positive association was observed for both bacterial and viral infections, and especially for respiratory and skin infections. A dose-response relationship was also noted between the number of hospital-treated infections and the risk of HL.

EBV is the first human tumor virus discovered to be associated with multiple human malignancies, contributing to around 1.8% of all cancer cases across the world [5]. As lymphocytes, especially B cells, are the primary targets of EBV infection, the most common type of EBV-related malignancies is HL, Burkitt lymphoma [20,21], and T/NK-cell lymphomas [21]. In addition to hematological malignancies, EBV is also known to be associated with epithelial malignancies, especially non-keratinizing NPC [5]. Although the link between EBV and different malignancies has been well established, the underlying mechanisms remain poorly understood [22]. As only a few of the individuals infected with EBV eventually develop a malignant disease, other component causes play a pivotal role in the completion of the EBV-based causes of these malignancies. For instance, it has been suggested that NPC might be caused by complex interactions between genetic factors, non-genetic factors, and EBV infection [22].

Among non-genetic factors, two Nordic studies have related previous infectious and inflammatory diseases [23] and season of birth—as a proxy for infancy exposure to infections [24] and the risk of HL. In a population-based case-control study of NPC in China, we showed earlier that a history of chronic infection, e.g., sinusitis and otitis, was associated with a higher risk of NPC [25]. The present finding of a positive association between hospital-treated infections and a higher subsequent risk of HL and NPC, in a dose-response fashion, lends, therefore, additional support to infections, independently of or jointly with EBV as risk factors for HL and NPC.

Due to shared immune pathways in response to non-oncogenic and oncogenic infectious agents [12,26], the pathological behaviors of oncogenic agents might be enhanced (or hampered) by the presence of non-oncogenic infectious organisms via cross-immunity [12]. The interactions between immune pathways involved in protection against infections in general and those involved in protection against a specific oncogenic agent [12,26] might also contribute, as some infectious organisms may interfere with the transmission of oncogenic agents through immunological alterations [12]. For example, people with a latent EBV infection might be at higher risk of other infections [12]. Individuals with such increased risk due to altered host immunity might be a high-risk group that will go on to develop a malignant disease [12]. Regardless of EBV, disequilibrium in the immune-response homeostasis after an infectious event may disrupt the immunosurveillance [13], which is key to the control of the transition from normal to malignant cells as well as the control of malignant cell behaviors [12]. Finally, although this hypothesis is relatively new and under-investigated in EBV-related malignancies, a previous infectious event requiring hospital treatment was indeed shown to be associated with the risk of other EBV-related diseases, including multiple sclerosis [27]. Similarly, future studies should also aim to test the same hypothesis in malignancies related to other viruses to assess whether a generic mechanism exists for all infection-related malignancies.

To our knowledge, our study is the first to assess the associations of hospital-treated infections, by type, site, age, and frequency, with the risk of two EBV-related malignancies. The main strengths are the population-based study, the large sample size, and the nested case-control design, which preserves the validity of the underlying cohort. The complete follow-up due to linkages to the national registers and the objective and prospective ascertainment of infections and malignancies minimized selection and measurement biases commonly existent in observational studies.

The main limitation of the present study is the lack of data to confirm the EBV-based etiology of HL and NPC. However, as the median age of HL diagnosis was 71 in the present study and it is estimated that 55% of patients with an HL diagnosed at 55 or above are likely attributable to EBV, whereas 80% of NPC in low incidence areas is estimated to be attributable to EBV [11], we consider a large proportion of the studied HL and NPC cases to be related to EBV. For the same reason, we did not study other malignancies potentially related to EBV, such as Burkitt lymphoma and gastric cancer, as the proportions of EBV-attributable cases are believably much lower among these malignancies in Sweden.

A second limitation is the potential for surveillance bias, i.e., individuals with an infectious event requiring hospital treatment might be more surveyed, leading to a higher-than-expected detection rate of malignancy and reverse causation, i.e., infections might be a result instead of a risk factor of HL or NPC. To address these concerns, we excluded infections experienced three years before the index date in the main analysis and during five years before the index date in a sensitivity analysis. Although a 5-year lag time is often used in studies of cancer risk factors, a 3-year lag time is likely sufficient for HL and NPC. For instance, elevated levels of antibodies associated with EBV lytic infection could be detected approximately 37 months before the clinical diagnosis of NPC [28]. Further, due to the incomplete coverage of inpatient care data before 1987 and the lack of outpatient care data before 2001 in the Swedish Patient Register, some individuals with hospital-treated infections might have been misclassified as not having an infection. Similarly, we only focused on infections that required hospital treatment; study participants classified as unexposed might have been infected although not treated for such infection in the hospital. The observed associations may therefore be an underestimate of the real associations.

## 5. Conclusions

In conclusion, hospital-treated infections were associated with a higher subsequent risk of HL and NPC. These findings suggest that infectious events might contribute to the carcinogenesis of malignancies potentially related to EBV. Due to the observational nature of the study, interpretation of the present findings as causal should, however, be made with caution before independent validations from regions of different prevalence of EBV infection and the burden of these malignancies are available.

## Figures and Tables

**Table 1 cancers-14-03804-t001:** Baseline characteristics of patients with EBV-related malignancies and their matched controls.

Characteristics	Hodgkin’s Lymphoma		Nasopharyngeal Carcinoma	
	Controls	Cases	Controls	Cases
*N*	4450	890	1530	306
Age at the index date ^1^, *N* (%)				
Median, year	71.3	71.2	66.6	66.5
<60 years	1080 (24.3)	216 (24.3)	490 (32.0)	98 (32.0)
≥60 years	3370 (75.7)	674 (75.7)	1040 (68.0)	208 (68.0)
Sex, *N* (%)				
Male	2640 (59.3)	528 (59.3)	1110 (72.5)	222 (72.5)
Female	1810 (40.7)	362 (40.7)	420 (27.5)	84 (27.5)
Area of residence, *N* (%)				
Southern Sweden	1079 (24.2)	232 (26.1)	382 (25.0)	78 (25.5)
Central Sweden	2355 (52.9)	448 (50.3)	817 (53.4)	160 (52.3)
Northern Sweden	1016 (22.8)	210 (23.6)	331 (21.6)	68 (22.2)
Educational level, *N* (%)				
0–9 years	1893 (42.5)	400 (44.9)	552 (36.1)	144 (47.1)
10–12 years	1368 (30.7)	285 (32.0)	508 (33.2)	90 (29.4)
≥13 years	675 (15.2)	100 (11.2)	278 (18.2)	35 (11.4)
Missing	514 (11.6)	105 (11.8)	192 (12.5)	37 (12.1)
Family history, *N* (%)	1 (0.02)	6 (0.7)	1 (0.07)	1 (0.3)

^1^ Index date: date of diagnosis for cases and date of selection for controls.

**Table 2 cancers-14-03804-t002:** Odds ratios (ORs) and 95% confidence intervals (CIs) of EBV-related malignancies in relation to hospital-treated infections >3 years before malignancy diagnosis.

Hospital-Treated Infections	Hodgkin Lymphoma			Nasopharyngeal Carcinoma		
	*N* of Controls	*N* of Cases	OR (95%CI)	*N* of Controls	*N* of Cases	OR (95%CI)
** *Any infection* **						
No	3454	625	Ref.	1232	230	Ref.
Yes	996	265	1.49 (1.26–1.75)	298	76	1.36 (1.01–1.83)
** *Type of infection* **						
Viral infection						
No	4127	793	Ref.	1432	281	Ref.
Yes	323	97	1.57 (1.23–2.00)	98	25	1.30 (0.81–2.10)
Bacterial infection						
No	3908	736	Ref.	1376	264	Ref.
Yes	542	154	1.52 (1.25–1.86)	154	42	1.39 (0.96–2.02)
Other infection						
No	4322	858	Ref.	1502	299	Ref.
Yes	128	32	1.28 (0.86–1.90)	28	7	1.21 (0.51–2.88)
** *Site of infection* **						
CNS infection						
No	4397	873	Ref.	1510	301	Ref.
Yes	53	17	1.57 (0.90–2.74)	20	5	1.17 (0.43–3.16)
Gastrointestinal infection						
No	4251	849	Ref.	1469	292	Ref.
Yes	199	41	1.02 (0.72–1.45)	61	14	1.13 (0.62–2.07)
Respiratory infection						
No	4104	785	Ref.	1420	280	Ref.
Yes	346	105	1.59 (1.26–2.01)	110	26	1.16 (0.73–1.83)
Genitourinary infection						
No	4350	877	Ref.	1502	298	Ref.
Yes	100	13	0.63 (0.35–1.13)	28	8	1.42 (0.64–3.18)
Skin infection						
No	4349	847	Ref.	1506	299	Ref.
Yes	101	43	2.23 (1.54–3.22)	24	7	1.56 (0.64–3.78)
** *Age at infection* **						
Below 40						
No	4183	826	Ref.	1438	281	Ref.
Yes	267	64	1.26 (0.93–1.72)	92	25	1.43 (0.84–2.42)
40 to below 60						
No	4106	789	Ref.	1426	280	Ref.
Yes	344	101	1.55 (1.22–1.98)	104	26	1.27 (0.80–2.02)
60 or above						
No	3973	764	Ref.	1408	276	Ref.
Yes	477	126	1.42 (1.13–1.78)	122	30	1.28 (0.82–2.00)
** *Number of infections* **						
0	3454	625	Ref.	1232	230	Ref.
1	656	158	1.34 (1.10–1.64)	201	52	1.37 (0.97–1.93)
2	170	50	1.66 (1.19–2.32)	57	12	1.14 (0.58–2.24)
3 or more	170	57	1.89 (1.37–2.60)	40	12	1.56 (0.79–3.09)

Analyses adjusted for age, sex, calendar period, area of residence, and educational attainment.

**Table 3 cancers-14-03804-t003:** Odds ratios (ORs) and 95% confidence intervals (CIs) of EBV-related malignancies in relation to hospital-treated infections >3 years before malignancy diagnosis, stratified analysis by sex and age.

Characteristics	Hodgkin’s Lymphoma			Nasopharyngeal Carcinoma		
	*N* of Controls	*N* of Cases	OR (95%CI)	*N* of Controls	*N* of Cases	OR (95%CI)
**Male**						
No infection	2101	376	Ref.	912	174	Ref.
Any infection	539	152	1.60 (1.29–1.99)	198	48	1.24 (0.86–1.78)
**Female**						
No infection	1353	249	Ref.	320	56	Ref.
Any infection	457	113	1.35 (1.05–1.74)	100	28	1.69 (0.99–2.88)
**<60 at the index date**						
No infection	846	148	Ref.	394	75	Ref.
Any infection	234	68	1.75 (1.26–2.44)	96	23	1.22 (0.70–2.13)
**60 or above at the index date**						
No infection	2608	477	Ref.	838	155	Ref.
Any infection	762	197	1.43 (1.19–1.73)	202	53	1.44 (1.01–2.05)

Analyses adjusted for age, sex, calendar period, area of residence, and educational attainment.

## Data Availability

The data generated in this study are not publicly available for legal and ethical reasons. Data access is however possible upon request to the corresponding authors.

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
