# Peer review of "Hospital-Treated Infections and Increased Risk of Two EBV-Related Malignancies: A Nested Case-Control Study"

_cancers, 2022, doi:10.3390/cancers14153804_

Round 1
Reviewer 1 Report
This is an unusual and very interesting study based on abundant data gained from well documented registries.
There are only few minor points that should be considered:
- - Simple summary should be rewritten and both message that it sends and grammar should be considered.
- - Are any multiple-comparison correction methods used?
- - Where both HL and NPC diagnoses based on primary tumour tissue? There might be a difference between HL and NPC as NPC is usually diagnosed based on metastatic tissue that usually has EBV infection.
- - It would be beneficial to discuss why the authors expect prior non-EBV infections to promote EBV infection or EBV-induced/associated tumorigenesis. Is this a mechanism that could be applied to all virus-associated malignancies?
Reviewer 2 Report
I am glad to be able to read your large-scaled study discussing hospital EBV infection. Here are my recommendations as follows;
1. Are there any cases that developed both HL and NPC together?
2. Is there any other chance patients could be infected outside the hospital?
3. How can you suggest that hospitalization made patients infected by EBV mostly? You should inform the exact EBV infection rate of the hospital you investigated in Sweden at first.
Round 2
Reviewer 2 Report
"Hospital-treated infection" seems to be a very helpful expression!
